# Preparation of a Novel Flame Retardant Formulation for Cotton Fabric

**DOI:** 10.3390/ma13010054

**Published:** 2019-12-20

**Authors:** Hung Kim Nguyen, Wataru Sakai, Congtranh Nguyen

**Affiliations:** 1Department of Polymer Chemistry, Faculty of Chemistry, University of Science, Viet Nam National University of Ho Chi Minh City, Ho Chi Minh City 700000, Vietnam; nkhung@hcmus.edu.vn; 2Faculty of Materials Science and Engineering, Kyoto Institute of Technology, Matsugasaki, Sakyo, Kyoto 606-8585, Japan; wsakai@kit.ac.jp

**Keywords:** organophosphorus compounds, flame retardant, cotton fabrics, condensed phase

## Abstract

A novel halogen-free flame-retardant formulation was prepared and coated onto cotton fabrics. The structure of phosphorus compounds in the system was characterized by attenuated total reflectance Fourier transform infrared spectroscopy (ATR-FTIR) and nuclear magnetic resonance spectroscopy (^1^H-NMR). Results from the ATR-FTIR spectroscopy, scanning electron microscopy (SEM), and energy-dispersive X-ray spectroscopy (EDX) analyses presented that the flame retardant was coated successfully onto a cotton surface. We investigated the thermal stability and fire-retardant behaviors of cotton fabrics using thermal gravimetric analysis (TGA) and the vertical flame test. We also discuss the mechanism of flame retardance of coated cotton fabrics.

## 1. Introduction

Cotton fabrics are known as one of the most important natural fibers found in apparel production, home furnishings, and industrial product applications. This is due to cotton’s excellent properties such as biodegradability, softness, warmth, and recyclability [1]. However, the high flammability of cotton fabrics is a consideration in cotton-based productions [1,2]. A proper flame retardant (FR) should be applied to retard the ignition of the cotton fabrics and/or decrease flame spread [02]. Among the various technological methods developed by academics to prepare the durable FR finishing of cotton fabrics, FR coatings have become one of the most convenient, economical, and most efficient ways [3]. Cotton fabrics are often treated with reactive FR or they are back coated to improve flame retardancy behavior. For example, silicones [4,5], polyurethanes [6], and polyphosphonate (PEPBP) [7,8] coated onto cotton surfaces are utilized to provide adequate flame retardancy under adverse environmental conditions. In addition to polymer matrix-based FR, various active FRs were proposed to impart flame retardancy to cotton fabrics [9]. Given environmental concerns related to halogen FRs, phosphorus-based FRs are considered the most promising candidate because of their ecological properties and fire retardance in both the gas phase and condensed phase [10,11,12]. The recent researches have found the condensed-phase effectiveness in the preparation of phosphorus-based FR coatings or back coated for the cotton fabric process [13,14,15,16]. The FR compounds above may also contain phosphorus elements or combinations of P and N elements for synergistic interaction. The cross-linkable organophosphorus FR system was introduced into cotton fabrics. From the cotton or cotton/nylon blends treated with commercial FRs, oligomeric OH terminated methylphosphonate-phosphate has exhibited a highe char residue due to the catalysis effect of phosphoric acid on the cotton/nylon fabrics for the dehydration of cellulose. The char length of the treated cotton or treated cotton/nylon blends decreases compared to the length of the untreated cotton or single-fiber nylon fabrics [17]. 

Recently, multifunctional FR monomers containing triazine rings [18,19] were synthesized and introduced onto the cotton fabrics. Triazine-based and UV-curable flame retardants were also integrated as alternative durable FRs for cotton fabrics through polymerization under UV radiation [20]. C–O bonding was formed through active chlorine atoms in the triazine-based FRs owing to the hydroxyl bonding in the cellulose units. It was found that the durability of flame resistance of cotton fabric that diminished with triazine-based FRs may be improved significantly after some soaping cycles.

Other phosphorus-based FRs reported in the literature include phosphorus-containing acrylates, such as methacryloloxyethylorthophosphor tetraethyl diamidate (MPD), pyrovatex, tri(acryloyloxyethyl) phosphate, and triglycidyl isocyanurate acrylates. Acrylate FRs are introduced to the substrate through the UV curing or impregnation method. It was found that a chemical cross-linking reaction occurred between the FRs and cotton matrix, revealing excellent thermal stability under high-temperature conditions. Moreover, the FR behavior of the treated cotton indicated higher initial degradation temperature and more char yield compared to that of the neat cotton.

To develop novel phosphorus-containing FRs for cotton fabrics, this study synthesized a new FR formulation using anhydride methacrylic and bis(hydroxymethyl)phosphinic acid. The formulation’s structure was characterized using FTIR and ^1^H NMR analysis. The surface and thermal behaviors of the treated cotton were characterized using ATR-FTIR, SEM, EDX, and TGA analysis. Furthermore, we proposed the FR mechanism of the organophosphorus formulation on cotton fabrics by investigating the chemical structure and elemental compositions of the char residues.

## 2. Experimental

### 2.1. Materials

Cotton fabric: 100% scoured and bleached plain-weave cotton fabrics were purchased from commercial entities in Vietnam with a density of 237 g/m^2^.

Methacrylic anhydride (MAAH) was purchased from Sigma-Aldrich Co. Munich, Germany. Benzoyl peroxide (BPO) was purchased from Merck Co. Darmstadt, Germany. Triethylamine (TEA), acetone, 37% HCl solution, and 25% NH_3_ solution were purchased from Xilong Chemical Co., shantou, China. All reagents were used without any further purification.

### 2.2. Preparation of the Phosphorus Containing Formulation 

Bis(hydroxymethyl)phosphinic- methacrylate (PMA) system was synthesized through the two-step reaction presented in Scheme 1.

Starting bis(hydroxymethyl)phosphinic acid (BHMP) was not commercial, and the authors prepared it according to the protocol described in Reference [21]. BHMP (10.08 g, 0.08 mol) and triethylamine (8.08 g, 0.08 mol) were placed into a three neck-round-bottom flask equipped with a temperature controller, a reflux condenser, and a mechanical stirrer. The temperature of the flask was maintained at 0−5 °C. Methacrylic anhydride (24.64 g, 0.16 mol) was placed in an addition funnel, and it was added slowly to the flask containing the BHMP solution over an hour. Then, the reaction mixture was stirred at 40 °C for 24 h. A yellowish product was obtained with a yield of 90%. 

### 2.3. Fabric Treatment

For irradiation, we used a metal halide lamp (made in China) with a broad-band UV-source. Blank cotton fabrics were immersed in the acetone solution containing the FR system (as shown in Table 1) in the presence of 1% BPO at room temperature and then neutralized by ammonia solution to pH 7–8. After 30 min, the impregnated fabric was removed from the solution, placed on a glass plate, dried at 90 °C for 3 min, and then cured by UV radiation on both sides for 5 min each. After radiation, the fabric was soaked with water and then sequence impregnated in 0.5% HCl solution and 1% NH_3_ solution for 3 min. Finally, after washing ten times, each sample was allowed to air dry at room temperature (30 °C) until no weight loss was detected.

### 2.4. Measurements

#### 2.4.1. FTIR and ^1^H-NMR 

FTIR spectroscopy was carried out on a Jasco FT/IR 4700, Kyoto, Japan to characterize the structure of the PMA using a thin KBr disk. The surface functional groups of the samples were investigated by ATR-FTIR using a diamond crystal at 32 scans (Jasco FT/IR 4700). The measurement was carried out in the range of 4000–500 cm^−1^ by 32 scans, where the resolution was 4 cm^−1^.

The ^1^H-NMR measurement was recorded on a Bruker AV 500 MHz spectrometer, Ho Chi Minh City, Vietnam, in DMSO, with tetramethylsilane (TMS) as the reference.

#### 2.4.2. Weight Gain

All cotton fabrics were dried in an oven at 90 °C for 30 min, and then weighed quickly. The weight gain of the treated fabric was obtained using the following equation:
Weight gain (%) = 100 × (W_2_ − W_1_)/W_1_(1)
where W_1_ and W_2_ represent the weights of the untreated fabrics and treated ones, respectively.

#### 2.4.3. Scanning Electron Microscopy (SEM)

The surface morphology and chemical compositions of the fabric samples were acquired using SEM (Hitachi S-3000N, Kyoto, Japan) equipped with EDX spectrometers. The specimens has been coated with a conductive layer.

#### 2.4.4. Vertical Flame Test

The vertical flame test was carried out according to the DIN 53906 standard method [22]. The sample size was 150 mm × 75 mm. Butane gas was selected for the combustion. The flame height and burning time were about 40 mm and 10 s, respectively. The average flaming time (for both the after flame and afterglow) of the five test specimens was recorded. 

#### 2.4.5. Thermogravimetric Analysis (TGA)

To characterize the thermal properties of the cotton fabrics, we employed thermogravimetric analysis (Discovery TA Instrument, Kyoto, Japan) under nitrogen and air atmospheres from 30 to 600 °C at a heating rate of 10 °C /min. 

#### 2.4.6. Durability Test

The flame durability test was evaluated using the ISO 105-C10:2006 standard [23]. The laundering process used a non-ionic surfactant. The temperature of the laundering solution was kept at approximately 40 °C. Each cotton sample was washed ten times continuously.

## 3. Results and Discussion

### 3.1. Synthesis of the PMA

A novel FR formulation was prepared using the two-step reaction presented in Scheme 1. Figure 1 shows the FTIR spectrum of the phosphorus-containing formulations. The absorption bands of the saturation carbon (sp^3^) were observed at 2988–2900 and 1430 cm^−1^. The peak at 3363 cm^−1^ contributed to the stretching vibration of the –OH group. The peaks at 1720 and 1626 cm^−1^ corresponded to the vibration of the C=O and C=C groups, respectively. The peaks at 1168 and 1041 cm^−1^ could be assigned to the stretching band of P=O and P–O–C bonding, respectively.

Figure 2 shows the ^1^H-NMR spectrum of the phosphorus-containing formulation. The chemical shifts of protons were observed at 1.80−1.86 (s, CH_3_–C), 3,59 (d, P–CH_2_–OH), 4.20–4.25 (d, P–CH_2_–OC(O)), and 5.60–6.05 (d, CH_2_=CH–). The observed signals at 4.20 (d) and 4.25 (d), respectively, corresponded to the chemical shifts of the protons in P–CH_2_ in the BMMP. Table 2 shows the peak assignments of three UV curable monomers (BMMP, HMMP, and MA) indicated in the FR formulation. The FTIR and ^1^H–NMR results indicated that the novel FR formulation was prepared successfully, and it was able to be used as a UV curable FR formulation without further purification.

### 3.2. Surface Characterization of the Treated Cotton Fabrics

We investigated the fabric surface properties using ATR-FTIR and SEM image analyses. Figure 3 shows the ATR-FTIR spectrum of untreated (COT) and FR coated cotton fabrics (COT-30). The results showed that both COT and COT-30 fabrics displayed stretching vibration modes of –OH, C–H (sp^3^), and C–O groups in cellulose at 3300, 2950, and 1050 cm^−1^, respectively. Compared to the ART-FTIR spectrum of the uncoated COT, two new absorption appearances at 1710 cm^−1^ and 1546 cm^−1^ (Figure 3b) were attributed to –COO–CH_2_–P (ester) and –COO– (carboxylate anion of the ammonium salt), respectively. These results concealed that the FRs were coated on the cotton surfaces. 

The surface morphology structure of untreated and PMA-treated cotton fabrics was also characterized using SEM analysis. The smooth fabric surface was observed in the SEM image of the neat cotton fabrics (Figure 4A,A1). Meanwhile, the PMA-treated cotton fabrics displayed rougher and more intact features with many layers of polymer infiltrating to the cotton fabrics (Figure 4B–D). We observed that a lot of the PMA polymers covered the fibers in the COT-20, COT-25, and COT-30 fabrics. Denser layers were observed as the loadings of the FR increased. 

To further support the above assertion, the elemental compositions of the untreated and treated cotton fabrics were analyzed using EDX experiments. Figure 5 shows the EDX results of the COT, COT-20, COT-25, and COT-30 samples. We found there was limited P content (0.04% only) and low N content (3.67%) due to the non-cellulose impurities showing in fibers [01] in the COT sample. We found that changes in the P and N contents were determined from 0.23% to 0.60% and 6.06% to 6.50% as the weight loadings of FRs added increased from 20% to 30%, respectively. The noticeable increase in the P and N contents of the treated cotton fabrics was revealed compared to that of the untreated cotton fabrics. Moreover, the P and N contents on the treated cotton fabrics increased with an increasing PMA concentration in the finishing solution. From Figure 6, the data from the P and N element mapping images of the treated cotton fabrics showed that the P and N elements were almost uniformly distributed in all the treated cotton fabrics. The higher the weight loadings of the FR, the denser the density of distribution. Despite all the PMA-treated cotton fabrics being washed ten times with the soaping solution, the FRs were still retained on the cotton layers in all the PMA-treated cotton fabrics. Therefore, we concluded from these findings that the FR formulation was coated onto the surface of the cotton fabrics.

### 3.3. Flame Retardant Performance

We investigated the flame retarding performances of the treated cotton fabrics using the vertical flame test. Following DIN 53906, we recorded the average times for both the after flame and afterglow of each of the five test samples. The flammability results of the cotton fabrics and images taken after the vertical burning tests are shown in Table 3. The neat cotton fabrics were natural to burn and were almost destroyed without any char residues remaining. The COT sample burned out violently with after flame and afterglow times of 18 s and 14 s, respectively. On the contrary, the COT-A fabrics without phosphorous-containing FRs behaved as highly flammable fabrics, and no char residues were obtained after the combustion. However, a very thin and light char was formed during the COT-A combustion. This showed that ammonium carboxylate decomposition generated ammonia gas and carboxylic acid groups during the flaming. These carboxylic acid groups accelerated the dehydration of the cotton fabrics. We observed that increasingly dehydrated cotton formed thermally stable carbon-rich residues. The ammoniac release gas might dilute the vapor phase gas on the cotton surface. On the other hand, this diluted vapor phase gas partially retarded the complete degradation of the cotton into CO_2_ and H_2_O, though negligible [24].

Table 3 shows that the inflammability behaviors of the treated cotton fabrics were enhanced as more FRs were added. We noted that both the after time and afterglow times of the coated COT-fabrics were achieved at lower values compared to the values of the neat cotton or COT-A. Both COT-25 and COT-30 fabrics showed self-extinguishing with no after time and afterglow time observed after applying the 10 s burning test. In particular, both COT-25 and COT-30 fabrics exhibited char length lower than 150 mm, thereby passing the DIN 53906 standard for both COT-25 and COT-30 fabrics. Meanwhile, the higher after flame time of COT-20 compared to that of neat cotton was 25 s, while the afterglow time of the COT-20 fabrics was zero. This result indicated that the char formed from COT-20 burning was thermally stable. Since the char length reached over 150 mm, the COT-20 sample did not pass the DIN 53906 standard. It is considered that the char forming ability of UV curable BMMP and HMMP FRs were responsible for the improvement in flame retardancy on the cotton fabrics.

To understand the flame retardancy behaviors of the UV curable FRs, we investigated the thermal stabilities of the neat cotton, coated cotton fabrics, and PMA homopolymer using TGA analysis. All the TGA curves and thermal analysis data of the samples carried out under both nitrogen and oxygen atmosphere are shown in Figure 7, Figure 8, and Table 4, respectively. All TGA curves exhibited at least two main stages as the temperature increased from 50 °C to 600 °C with a heating rate of 10 °C /min. The 5 wt% of weight loss around 100−120 °C for all samples in the first stage corresponded to the evaporation of water due to moisture absorption. The primary decomposition stage with the highest-rated and highest weight loss (87%) at 330–400 °C was attributed to hydration and further cellulose degradation in the cotton [25]. For the FR coated cotton fabrics, we observed the same thermal degradation at a similar temperature range (260–350 °C) even though the weight loss was not significant (45–50%) compared to that of the neat cotton fabrics. The degradation appearing at lower temperatures (about 260 °C for FR coated COT samples and 300 °C for PMA homopolymer) was related to the formation of phosphoric acid derivatives from the decomposition of the FR, the cleavage of the aliphatic fraction, and the cellulose dehydration of the cotton fabrics [25]. The gradual degradation of both the FR coated cotton samples and PMA homopolymer appearing at 340–430 °C was related to network formation through the esterification of phosphinic acids. A large amount of charred residue was observed for all the FR coated cottons (about 27.0, 27.8, 30.1, and 35.2% in PMA homopolymer, COT-20, COT-25, and COT-30, respectively), while very little char (7.1%) was recorded in the neat cotton (Figure 7). Another noticeable thing was that all TGA curves of the samples in an oxygen atmosphere showed the same degradation behavior beyond 300 °C compared to the curves in the nitrogen atmosphere (Figure 9). However, a lower amount of char residues in the oxygen gas was obtained at 600 °C for both the neat cotton fabrics and FR coated cotton fabrics compared to the residues in the nitrogen medium. That is, 0.2 % for the COT and COT-A, and about 27.0, 27.8, 30.1, and 35.2% in PMA homopolymer, COT-20, COT-25, and COT-30, respectively. The lower amount resulted from these char residues being partially volatilized because of the further oxidation. With little char residue during the COT-A combustion, polymethacrylic coated cotton fabrics showed no contribution to enhancing the thermal stability behavior of the cotton fabrics. On the contrary, the formation of significant char residues through the further reaction of cotton with the phosphorus-containing FRs (COT-20, COT-25, and COT-30) was responsible for the flame inhibition action. Therefore, we concluded that the compound containing phosphinic moiety played a role in the efficient char-forming FR of cotton fabrics in the condensed phase. However, its initial degradation occurred at quite lower temperatures.

To fortify for the above assertion, the charred fabrics were characterized by ATR-FTIR, SEM, and EDX analyses. Comparison of the ATR-FTIR spectra of COT-30 (a) and COT-30-char at 600 ^o^C (b) (Figure 10) revealed that the formation of unsaturated carbon–carbon bonds and the phosphorus containing charred residues occurred at elevated temperatures. The disappearance of the vibration at 2950 cm^−1^ during heating confirmed the loss of –CH_2_– groups. The formation of a new peak at around 1689 and 1575 cm^−1^, corresponding to the vibration modes of C=C bonds in the aromatic group, indicated the formation of graphite [26]. Moreover, the significant peak at 1193 cm^−1^ (medium) was assigned to the vibration of P=O bonds in the phosphate moieties. Therefore, we concluded from these findings that phosphorylated cellulose occurred under thermal degradation to promote thermally stable residue formation. Similar findings were concluded by Gaan et al. [27] for the actions of various phosphonate derivatives on cotton fabrics.

We also consolidated other evidence from the EDX analysis and SEM images of char residues of COT-20, COT-25, and COT-30. SEM images of the surface morphology of char residues (Figure 11) displayed the morphology of the cotton coated with different FRs loadings. Extremely thin char fibers were obtained for all samples. The pieces of cotton covered with lower FR loadings tended to produce fiber clusters and became fiber chars (Figure 11A). The fiber clusters moved closer, and the fiber char became bigger with higher FR loadings (Figure 11B). The addition of FR to cotton fabrics resulted in increased fiber chars. The most abundant fibers arranged close together were found in the char fibers of COT-30 (Figure 11C). Figure 11A2,B2,C2 show that the size of the char fibers increased when the loadings of the FR coated on the cotton increased. 

Moreover, all EDX results presented in Figure 12 show that the char residues had very high carbon contents. We believed that the structure of the char fiber was mainly carbon backbone retained by the FRs. Additionally, the P contents in the treated fabrics before and after burning were also considered (Figure 13). We found that the P contents in the char residues were higher than the unburned fabric. Furthermore, the distribution of P elements became denser, and the density of P elements in the char residue increased dramatically with the increase in FR loadings (Figure 14) compared to the cotton fabrics without heating treatment. 

## 4. Conclusions

A novel FR formulation was synthesized successfully using a simple process. The precise structure of FRs also was elucidated by FTIR and ^1^H-NMR spectroscopy analyses. The results of the ATR-FTIR, SEM, and EDX study showed that the PMA system monomer was coated onto the surface of the cotton fabrics. FRs promoted thermally stable char formation. The thick char fibers containing a uniform distribution of P elements were obtained from the FR coated cotton samples. The thermal stability of the PMA coated cotton fabrics was significantly enhanced. The PMA system used as a durable flame retardant exhibited high effectiveness of the condensed phase for cotton fabrics. We observed that the system passed the DIN 53906 standard with a 25% add-on.

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
