# Peer review of "Preparation of a Novel Flame Retardant Formulation for Cotton Fabric"

_materials, 2019, doi:10.3390/ma13010054_

Round 1
Reviewer 1 Report
The submitted manuscript is very interesting. Unfortunately, there are many errors to be addressed. First of all the language and grammar is not at expected level. There are many awkwardly used words or phrases especially in the Result and discussion section. As a rule, authors used too large and complicated sentences which possibly may be confusing for the average reader. Authors are advised to rephrase this long and complicated sentences into smaller ones from which it will be easy to distinguish what they intended to emphasize. Furthermore, in case of reproducibility authors didn’t stressed out how many (number) samples of each cotton sample they used in investigation. In Fig. 1 authors showed FTIR spectrum of PMA. Why didn’t they mark the peak at 1430 cm-1 as stated in the text? Likewise, on Fig. 3 they showed ATR-FTIR spectrum of COT (a) and COT-30 (b). What about the samples COT20 and COT25? Why they aren’t showed and discussed as well? Fig. 4 resolution is low. Authors need to emphasize that Figs A1, B1, C1 and D1 are same as corresponding ones but with larger magnification. TGA analysis was used to “To understand better the flame retardancy behaviors of UV curable FRs,…”. According to Table 4 and Figs. 7 and 8 it seems that samples COT20, COT25 and COT30 are showing inferior thermal stability (Onset) than sample COT; at least in temperature region 30 – 350 °C. Authors concluded that “…weight loss around 100-120 °C for all samples in the first stage corresponds to the evaporation of water due to the moisture absorption.” In section 2.3. Fabric treatment authors described that “Finally, after washing ten times, each sample was dried in air until no weight loss could be detected.” Here more data are needed to clarify at what temperature drying was carried out. Evidently the water didn’t evaporate; even though authors dried the samples until no weight loss could be detected. Furthermore, Table 4 need to be revised, especially the column 3 (Char at 600 °C in Nitrogen); these data are not logical. The sentence “Another noticeable thing is that all TGA curves of these samples in an oxygen atmosphere showed no different degradation beyond 300 °C as compared to that in the nitrogen atmosphere.” need to be revised and explained. As a part of the solution of the latter, authors are advised to include the one additional figure in this manuscript which will more clearly show the comparison of TG degradation in nitrogen and oxygen; not necessarily for all samples.
Author Response
The submitted manuscript is very interesting. Unfortunately, there are many errors to be addressed. First of all the language and grammar is not at expected level. There are many awkwardly used words or phrases especially in the Result and discussion section. As a rule, authors used too large and complicated sentences which possibly may be confusing for the average reader. Authors are advised to rephrase this long and complicated sentences into smaller ones from which it will be easy to distinguish what they intended to emphasize.
Our answer: Thanks for your appreciation and offer. The language and grammar have been corrected.
In case of reproducibility authors didn’t stressed out how many (number) samples of each cotton sample they used in investigation. In Fig. 1 authors showed FTIR spectrum of PMA. Why didn’t they mark the peak at 1430 cm-1 as stated in the text? Likewise, on Fig. 3 they showed ATR-FTIR spectrum of COT (a) and COT-30 (b). What about the samples COT20 and COT25? Why they aren’t showed and discussed as well?
Our answer: thank you very much for bringing this inconsistency to our attention.
Indeed, only one representative FR coated cotton fabric (COT-30) was chosen to report and discuss. And FTIR spectra of COT20 and COT25 also showed the same vibrations as COT-30’s.
Fig. 4 resolution is low. Authors need to emphasize that Figs A1, B1, C1 and D1 are same as corresponding ones but with larger magnification.
Our answer: Thank you for this comment, we have adjusted all them and all SEM are replaced.
TGA analysis was used to “To understand better the flame retardancy behaviors of UV curable FRs,…”. According to Table 4 and Figs. 7 and 8 it seems that samples COT20, COT25 and COT30 are showing inferior thermal stability (Onset) than sample COT; at least in temperature region 30 – 350 °C. Authors concluded that “…weight loss around 100-120 °C for all samples in the first stage corresponds to the evaporation of water due to the moisture absorption.” In section 2.3. Fabric treatment authors described that “Finally, after washing ten times, each sample was dried in air until no weight loss could be detected.” Here more data are needed to clarify at what temperature drying was carried out. Evidently the water didn’t evaporate; even though authors dried the samples until no weight loss could be detected.
Our answer: thank you for pointing this out.
All samples COT20, COT25, and COT30 showed inferior thermal stability (Onset temp.) than sample COT (250-350oC). This early degradation was related to the formation of phosphoric acid derivatives results of the decomposition of the FRs, and the cleavage of the aliphatic fraction of FRs coated on the cotton fabrics. The cause of this may be the acceleration of cellulose dehydration.
After handling the FR, all cotton fabrics were washed ten times, and each sample was allowed under the air dry at room temperature (30 oC) until no weight loss was detected. And the first steps with a five percentages of weight loss around 100-120 °C for all samples was suggested to the evaporation of water due to the moisture absorption.
Furthermore, Table 4 need to be revised, especially the column 3 (Char at 600 °C in Nitrogen); these data are not logical.
Our answer: thank you for this point. All corrected data were added.
The sentence “Another noticeable thing is that all TGA curves of these samples in an oxygen atmosphere showed no different degradation beyond 300 °C as compared to that in the nitrogen atmosphere.” need to be revised and explained. As a part of the solution of the latter, authors are advised to include the one additional figure in this manuscript which will more clearly show the comparison of TG degradation in nitrogen and oxygen; not necessarily for all samples.
Our answer: Thank you for raising this point. We have reformulated the statement to read, as indicated below. And, Fig 9. TGA curves of COT-30 in nitrogen and oxygen atmosphere have been added.
“Another noticeable thing is that all TGA curves of these samples in an oxygen atmosphere showed the same degradation behavior beyond 300 oC as compared to these in the nitrogen atmosphere (Fig. 9).

Reviewer 2 Report
Reviewer’s comment
Manuscript Number: materials-641549
The authors reported phosphorus-based flame retardant (FR) synthesised by using sodium hypophosphite, formaldehyde, and methacrylic anhydride (MAAH), which is applicable as a flame retardant for cotton fabric coating. The synthesis of PMA is novel but there are significant problems that should be tackled by the authors. This manuscript cannot be considered for publication in Materials until the following points are clearly addressed:
The PMA loading on cotton fabric to pass a vertical burning test (DIN 53906) is considerably higher compared to that of other phosphorus-based FR for cotton. The authors should highlight the novelty and significant contributions of PMA for flame retardant applications. Page 8: The nitrogen and phosphorus contents in PMA treated cotton fabric were increased with the increasing PMA loading. However, PMA has no nitrogen in its molecular structure. Why did the nitrogen content increase? Figure 11: The nitrogen content of PMA treated cotton fabric was not changed after the combustion. Why does the nitrogen remain in the char? Page 10, row 231: ‘Another noticeable thing is that all TGA curves of these samples in an oxygen atmosphere showed no different degradation beyond 300 oC as compared to that in the nitrogen atmosphere.’ is not correct. The authors should show DTA curves in Figures 7 and 8. The authors need to go through the manuscript carefully. There are some typos errors.Author Response
The PMA loading on cotton fabric to pass a vertical burning test (DIN 53906) is considerably higher compared to that of other phosphorus-based FR for cotton. The authors should highlight the novelty and significant contributions of PMA for flame retardant applications. Page 8: The nitrogen and phosphorus contents in PMA treated cotton fabric were increased with the increasing PMA loading. However, PMA has no nitrogen in its molecular structure. Why did the nitrogen content increase? Figure 11: The nitrogen content of PMA treated cotton fabric was not changed after the combustion. Why does the nitrogen remain in the char? Page 10, row 231: ‘Another noticeable thing is that all TGA curves of these samples in an oxygen atmosphere showed no different degradation beyond 300 oC as compared to that in the nitrogen atmosphere.’ is not correct. The authors should show DTA curves in Figures 7 and 8. The authors need to go through the manuscript carefully. There are some typos errors.
Our answer: The PMA solution (solvent) of the cotton impregnation was neutralized by ammonia solution to pH 7-8. Indeed, the PMA system consists of three monomers (BMMP, HMMP, and methylacrylic acid). There will be the formation of ammonium salt from the carboxylic group in the PMA system. And, the nitrogen-containing char residues was generated during the FR coated cotton fabrics burning. Besides, the neat cotton fabrics also produced the nitrogen char residue due to the impurities.

Reviewer 3 Report
Manuscript concerns synthesis and characterization of halogen-free flame retardant formulation. In vertical flame test, fire retardant behaviors of cotton fabrics coated with developed flame retardant formulation, was investigated. Results were compared with neat cotton, showing improvement of flame retardancy on the cotton coated with formulation.
Some points should be improved before publication. Details are listed below.
- line 203 – “flammability behaviors of treated cotton fabrics were enhanced as more FRs are added”- Do you mean flammability was reduced?
- Why there are such differences between afterflame time of COT-20 vs. COT-25 and COT-30.
- Did vertical flame test was repeated for each material?
- Make sure to use the same symbol for Celsius degree along the text, e.g. see line 218 and 219.
- Table 4 – How many times test was repeated for each sample (errors are missing)? Also, in what unit the values of char yield at 600°C were expressed?
- Figs. 4 and 10 – scale bars in SEM images are not clearly visible.
- Fig. 12 – error bars should be added.
- According to Instruction for Authors: “For research articles with several authors, a short paragraph specifying their individual contributions must be provided”.
Author Response
- line 203 – “flammability behaviors of treated cotton fabrics were enhanced as more FRs are added”- Do you mean flammability was reduced?
Our answer: thank you for this point. We have reformulated the statement to read, as indicated below.
“inflammability behaviors of treated cotton fabrics were enhanced as more FRs are added”
Our answer: thank you for this point.
- Why there are such differences between afterflame time of COT-20 vs. COT-25 and COT-30.
- Did vertical flame test was repeated for each material?
- Make sure to use the same symbol for Celsius degree along the text, e.g. see line 218 and 219.
- Table 4 – How many times test was repeated for each sample (errors are missing)? Also, in what unit the values of char yield at 600°C were expressed?
- Figs. 4 and 10 – scale bars in SEM images are not clearly visible.
- Fig. 12 – error bars should be added.
Our answer: Thank you for your kindly review.
- “Figs. 4 and 10 – scale bars in SEM images are not clearly visible and Fig. 12 – error bars should be added.” The scale bars in SEM images are added.
- Table 4 – How many times test was repeated for each sample (errors are missing)? Also, in what unit the values of char yield at 600°C were expressed?
Each test sample was repeatedly 5 times and the average time for both after flame and afterglow was reported. And, “The unit of char yield” (wt%) was added.
- According to Instruction for Authors: “For research articles with several authors, a short paragraph specifying their individual contributions must be provided”.
Our answer: Thank you for this point. We have reformulated the statement.

Round 2
Reviewer 2 Report
The authors have addressed most of my concerns. This manuscript can be recommed for publication after some typo errors (ex. lambs page3, row 86) correction.
Author Response
The authors have addressed most of my concerns. This manuscript can be recommed for publication after some typo errors (ex. lambs page3, row 86) correction.
Our answer: Thank you for this point. We have reformulated the statement to read.
"The metal halide lamb (made in China) with the broad-band UV-source was used for irradiation"
